# Bulged and Canonical G-Quadruplex Conformations Determine NDPK Binding Specificity

**DOI:** 10.3390/molecules24101988

**Published:** 2019-05-23

**Authors:** Mykhailo Kopylov, Trevia M. Jackson, M. Elizabeth Stroupe

**Affiliations:** 1Department of Biological Science and Institute of Molecular Biophysics, Florida State University, 91 Chieftain Way, Tallahassee, FL 32306, USA; mkopylov@nysbc.org (M.K.); tmj12b@my.fsu.edu (T.M.J.); 2New York Structural Biology Center, 89 Convent Ave, New York, NY 10027, USA

**Keywords:** G-quadruplex, G4, nucleoside diphosphate kinase, NDPK

## Abstract

Guanine-rich DNA strands can adopt tertiary structures known as G-quadruplexes (G4s) that form when Hoogsteen base-paired guanines assemble as planar stacks, stabilized by a central cation like K^+^. In this study, we investigated the conformational heterogeneity of a G-rich sequence from the 5′ untranslated region of the *Zea mays*
*hexokinase4* gene. This sequence adopted an extensively polymorphic G-quadruplex, including non-canonical bulged G-quadruplex folds that co-existed in solution. The nature of this polymorphism depended, in part, on the incorporation of different sets of adjacent guanines into a quadruplex core, which permitted the formation of the different conformations. Additionally, we showed that the maize homolog of the human nucleoside diphosphate kinase (NDPK) NM23-H2 protein—ZmNDPK1—specifically recognizes and promotes formation of a subset of these conformations. Heteromorphic G-quadruplexes play a role in microorganisms’ ability to evade the host immune system, so we also discuss how the underlying properties that determine heterogeneity of this sequence could apply to microorganism G4s.

## 1. Introduction

Traditionally, DNA is thought of as the genetic storage unit held in a double-stranded helical conformation. The famous double helix structure [1] falls short in light of the observations that guanine bases (Gs) in G-rich regions of DNA or RNA can form Hoogsteen base pairs with one another to create a planar G-quartet [2,3]. Sequential G-quartets can stack to form G-quadruplexes (G4s). In microorganisms, these secondary structures can play a role in antigenic variation to assist in evasion of the host immune response and in establishing viral latency [4]. G4s have been identified in eukaryotic nuclei using G4-specific antibody staining [5]. Further, functional roles in regulating transcription and replication continue to be identified from bacteria to mammals [6,7,8,9,10,11]. In short, DNA G4s are now a recognized structural form of DNA despite the initial controversy about their biological relevance.

G4s are identified throughout microorganisms—including viral, mammalian, and plant genomes—at similar, but not identical, loci. In bacteria, G4s are enriched in regulatory sequences as well as transfer, non-coding, and messenger RNAs [12]. In viruses, G4s that are conserved across viral classes are found in gene promoters and long terminal repeats, implicating them in gene expression regulation and viral latency [13,14]. In humans, G4s are enriched just upstream of transcription start sites (TSSs), as well as in introns near intron–exon boundaries, and are more commonly found in the sense strand and thus transcribed into mRNA [11]. Others are associated with telomeres [11] or oncogene promoters [15]. In the maize genome, G4s tend to occur just downstream of TSSs in the antisense strand (called “A5U”-type G4s for antisense 5′-untranslated region) and putative G4s are overrepresented in promoter regions of genes associated with energy status pathways, oxidative stress response, and hypoxia, suggesting a regulatory role for these elements [16]. For these reasons, G4 aptamer-based therapeutics that can inhibit bacteria-host cell interactions [17], override transcriptional [18] or epigenetic [19] signals, or regulate viral lifecycle [20] are an exciting avenue for drug discovery [21].

The analogy between these divergent genomes carries over into the surprising structural plasticity of the G4s and the protein factors that regulate G4 formation and dissolution in the nucleus and/or cytoplasm. For example, the ability for G4s in various pathogens to adopt protein-specific conformations suggests a role in those microorganisms to evade the host immune response [4,22]. Some viral genomes combine Watson–Crick base pairs with a G4 structure, perhaps influencing gene expression in HIV-1 [23]. Further, a number of mammalian proteins play a role in dissolving [24,25,26,27,28,29] or stabilizing [30,31,32] G4 interactions. Interestingly, we recently identified a nucleoside diphosphate kinase (NDPK) from maize, ZmNDPK1, that is analogous to the human NM23-H2, an NDPK homolog [33] (both bind the folded form of G4 DNA). However, the nature of their interaction is not completely understood because there is no reported three-dimensional structure of an NDPK–G4 DNA complex.

With accepting the importance of genic DNA G4s come questions about how they function. In this realm, biochemistry and biophysics shed light on the physical properties that lay the basis for defining these activities. To this end, various spectrophotometric and spectroscopic assays are available to assess the structure of model oligonucleotides in solution. Nonetheless, these techniques individually fall short of defining a single state of the oligonucleotide, and so must be used together to understand the G4 forming potential of any given G-rich stretch of nucleic acid. Structure determination is complicated by the innate polymorphism of even the simplest G4-forming sequences [34], formation of stable structures with bulges in G-tracts [35], topological interconversion [36,37], and G-tract slippage [38].

In this study, we combined UV spectrophotometry, CD spectrophotometry, and DMS footprinting to characterize the G4-forming DNA oligonucleotide *hex4*_A5U, which is derived from the 5′ untranslated region of the maize *hexokinase4* gene [16]. Each technique has different strengths in analyzing G4 structures, so we used them together to understand the biophysical properties of this sequence [39,40,41]. We then applied our analysis to characterizing the interaction between ZmNDPK1 and the G4 DNA, which form a high-affinity complex with a subset of the potential *hex4*_A5U G4 conformers [33]. Our analysis shows an unprecedented level of polymorphism in the *hex4*_A5U sequence that can be described by the topological isomers and G-register exchange concept [42], extended to include the non-canonical bulged G4 conformations. We further hypothesize that such polymorphism is a universal property of G4-forming sequences in eukaryotic as well as prokaryotic genomes.

## 2. Results

### 2.1. hex4_A5U Adopts a G4 Conformation in the Presence of Cations

We first tested the influence of different monovalent cations on the formation of G4 by the *hex4*_A5U sequence using UV-Vis and CD spectrophotometry. We will refer to the stretches of Gs as tracts I–IV, according to their order from the 56′ end of the DNA (Figure 1A). A characteristic G4 thermal difference spectrum (TDS) with a negative peak at 295 nm was obtained only in the presence of K^+^ ions (Figure 1B). TDS spectra of oligonucleotides annealed with Na^+^, Cs^+^, and Li^+^ ions were more negative at 295 nm than those determined in the absence of cations, but none had a prominent negative peak.

We next monitored the thermal denaturation of G4 structures by recording the change in absorbance at 295 nm (Figure 1C). A hypochromic shift at this wavelength with increasing temperature is associated with G4 melting [43,44]. In contrast, single-stranded DNA (ssDNA) experiences a hyperchromic shift at 295 nm upon increasing temperature due solely to denaturation of any transitory secondary structures such as ssDNA helix [45]. As expected, in the presence of K^+^ we observed a sigmoidal decrease in absorbance at 295 nm with increasing temperature, revealing a midpoint of transition (T*_1/2_*) at 58 °C. In the presence of Na^+^, Li^+^, or Cs^+^ ions, we saw an initial decrease in absorbance at 295 nm that suggested melting of a G4-like structure, followed by an increase in absorbance that we attributed to ssDNA helix denaturation. In the absence of cations, the absorbance at 295 nm steadily increased with increasing temperature. Further supporting our assessment that *hex4*_A5U forms a quadruplex structure, CD spectra characteristic of a parallel G4 conformation [46] were visible in K^+^, Na^+^, Li^+^, and Cs^+^ (Figure 1D). In the absence of any small cation, CD spectra indicate that the oligonucleotide is disordered (Figure 1D). Finally, analytical ultracentrifugation (AUC) showed that the DNA was folded into a compact globular structure with an average molecular weight of 10.8 kDa (expected: 10.2 kDa) and an average f/f0 of 1.56, indicative of a monomeric G4 (Figure 2).

The formation of G4-like structures in the presence of cations other than K^+^ was further evidenced by CD spectrophotometry. Samples annealed in the absence of cations had a positive peak maximum at 255 nm and did not undergo structural transitions with an increase in temperature (Figure 3A,B). At 25 °C, CD spectra of *hex4*_A5U annealed in the presence of any monovalent cation were similar to one another, whereas they were dramatically different from the spectra of *hex4*_A5U in the absence of cations. Negative ellipticity at 242 nm and positive ellipticity at 262 nm, as observed for cation-annealed samples, are the hallmarks of a parallel G4 [46]. K^+^-annealed samples melted as a single species with an increase in temperature (Figure 3C), whereas samples annealed in Na^+^, Li^+^, and Cs^+^ displayed a structural transition evidenced by a gradual shift of the maximum positive peak from 262 to 255 nm (Figure 3D–F).

### 2.2. hex4_A5U Oligonucleotide is a Mix of G4 Conformations

We performed DMS footprinting followed by piperidine cleavage (Figure 4) to identify the Gs involved in G4 formation in K^+^, and to characterize the G4-like structures that formed in the presence of non-K^+^ cations. G4 prediction by the Quadparser algorithm [49] flagged G_4_–G_6_, G_14_–G_16_, G_24_–G _26_, and G_28_–G _30_ as the four continuous G-tracts in the *hex4*_A5U sequence involved in G-tetrad formation [16]. A distinct footprinting pattern marked by missing products that correspond to Gs protected by G4 formation was seen only in K^+^-annealed samples (Figure 4A, lane 1). Specifically, bands corresponding to cleavage at G_3_–G_5_ (G-tract I from the Quadparser model), G_25_–G_26_ (partial G-tract III), and G_28_–G_30_ (G-tract IV) were missing, indicating that those Gs were strongly protected from being DMS-labeled. Low intensity bands corresponding to cleavage at G_6_ and G_24_ suggested weaker protection. In contrast, bands corresponding to cleavage at G_14_–G_16_ (G-tract II) and other discontinuous Gs (G_8_, G_11_, G_18_, G_19_, G_21_ and G_22_) were strong, indicating those Gs were not protected. Thus, we identified only two complete and one partial G-tract out of four G-tracts assigned by Quadparser for samples annealed in the presence of G4-inducing K^+^.

There was a similar, but much less prominent, footprinting pattern in the Na^+^, Li^+^, and Cs^+^-annealed samples (Figure 4A, lanes 2, 3, and 4) visible only in the 3′ region (compare intensity of the G_31_ band to G_28–_G_30_). In contrast, there was no protection in the absence of cations (Figure 4A, lane 5), and most of the oligonucleotide was degraded in the water alone (Figure 4A, lane 6). *hex4*_A5U DMS footprinting patterns in different cations agree with CD and UV-Vis melting experiments, which showed that K^+^, and to a lesser degree Na^+^, Li^+^, and Cs^+^, supported G4 formation, whereas no secondary structure was detectable in the absence of cations.

Despite unambiguous spectroscopic evidence of G4 formation in K^+^, our DMS footprinting failed to assign all G-tract II or III Gs (Figure 4A). This raised a question about the role of the middle Gs in the G4 structure, as well as the possibility of heterogeneous G4 structures. We first created a shorter construct, trim_A5U, with a three-base truncation at the 5′ end and a one-base truncation at the 3′ end, to simplify our analysis and eliminate the structures that would arise due to the G-register exchange (Figure 4B). All further analysis was done in trim_A5U background.

DMS footprinting of trim_A5U construct showed a less complicated footprinting pattern, where G_24_–G_25_ and G_28_–G_30_ at the 3′ end were clearly protected. Some difference in the degree of digestion was observed for G_25_ versus G_26_, G_18_ versus G_19_, and G_14_ versus G_15_–G_16_ (Figure 4B lane 2). Additionally, G_4_–G_6_ were less digested in KCl versus LiCl, indicating their involvement in G4 formation (Figure 4B, lanes 1 and 2). To verify that canonical G4 can still form, we further modified trim_A5U construct by replacing G_8_, G_18_, G_19_, G_21_, and G_22_ with thymidines, giving rise to a “locked” canonical construct—A5U^AH^. DMS footprinting of this locked variant clearly showed protection of 12 guanines in KCl but not LiCl. These guanines formed the G4 core, whereas overdigestion of G_11_ showed that it was not involved in core formation, as predicted.

Next, we employed rational mutagenesis to define the apparent heterogeneity of trim_A5U in G-tracts I, II, and III. Despite the predictions of middle G-tract involvement in G4 formation, deletion of the middle G-tract (G_14_–G_16_) or substitution of those Gs with adenines had no effect on G4 formation in the K^+^ assessed by TDS (Figure 5A). We already established that G_18_–G_19_ and G_21_–G_22_ could not be exclusively involved as a bulged G-tract II, since their substitution by thymidines resulted in a sequence that was still capable of G4 formation (Figure 4B and Figure 5A). Point mutations that simultaneously disrupted continuous G-tracts I, III, and IV (G_4_, G_25_, G_30_) resulted in a sequence that did not form a G4 (Figure 5B). To test the possibility of intermolecular G4 formation from a two-G-tract containing oligonucleotide, we made an A5U^R20^ (random 20) construct where the 5′ sequence upstream of the GGGAGGG hairpin was replaced with 20 random non-G bases. This oligonucleotide did not form a stable G4 (Figure 5B), although it had a CD spectrum indicative of parallel G4s when annealed with K^+^ or Li^+^ (Figure 5B).

From this analysis, we further hypothesized that the trim_A5U sequence exists as a mix of G4 conformers in G-tracts II and III, including variants where continuous G-tracts were interrupted by non-G bases, forming a bulge [35]. We called this central stretch of non-continuous Gs a “G-slide” region, which defines the G4 heterogeneity. According to this extended model, G-tracts I and IV are fixed, but G-tracts II and III are formed by six bases from a G-slide region of 10 Gs with or without one-base bulges (Figure 6A). Based on these assumptions we identified 13 possible variants (Figure 6B), named according to the participating G triplets that form G-tract II and III (A–H). To test our model, we designed “locked” sequences that enforced a single conformation (Figure 6B, Table 1).

### 2.3. Locked hex4_A5U Variants Form G4s with Distinct Properties

We proceeded to characterize the ability of locked trim_A5U variants to form G4s in K^+^ through our series of spectroscopic assays. TDS showed that all locked variants A5U^AD^–A5U^EH^ form G4s, albeit with variable amplitudes of the negative 295 nm peak (Figure 7A). CD spectra revealed additional differences between the variants (Figure 7B). Specifically, A5U^AD^, A5U^AH^, A5U^BH^, A5U^CF^, A5U^CH^, A5U^DH^, and A5U^EH^ had signatures of parallel G4s; A5U^AE^, A5U^AF^, A5U^AG^, A5U^BF^, and A5U^BG^ had signatures of antiparallel hybrid (anti-h) G4s; and A5U^CG^ had a mixed spectrum. Thermal denaturation experiments showed that A5U^AD^, A5U^CF^, and A5U^CG^ formed the weakest G4 structures, followed by A5U^BF^, A5U^BG^, and A5U^BH^, which together formed a group of G4 variants with *T_1/2_* <30 °C (Figure 7C). The remaining seven variants—A5U^AE^, A5U^AF^, A5U^AG^, A5U^AH^, A5U^CH^, A5U^DH^, and A5U^EH^—formed G4s that were stable at room temperature. All but A5U^AH^ contained one or two bulged G-tracts. Taken together, these data show that all 13 locked trim_A5U variants formed G4s but varied in their topology and thermal stability (Table 1).

### 2.4. ZmNDPK1 Requires Two Consecutive G-Tracts with a Single One-Base Loop for Efficient Binding

Previously, we demonstrated that ZmNDPK1 binds to wild-type *hex4*_A5U G4 DNA with high affinity [22]. We determined that ZmNDPK1 also binds with high affinity to trim_A5U (K_d_ = 16.6 nM), as well as to the locked canonical variant A5U^AH^ (K_d_ = 14.4 nM), but not to another locked variant A5U^AE^ (K_d_ = 194 nM) (Figure 8A–C). We further tested which locked variant competed with wild-type *hex4*_A5U for binding to ZmNDPK1 to assess its binding specificity. At a 100-fold excess of the competitor, locked variants showed varying degrees of competition efficiency (Figure 8D, Table 1). Although each of the 13 variants competed for binding, only those classified as parallel according to CD measurements competed with greater than 50% efficiency (Figure 8D). The common feature shared by the strong competitors (A5U^AD^, A5U^AH^, A5U^BH^, A5U^CF^, A5U^CH^, A5U^DH^, and A5U^EH^) was the presence of two G-tracts connected by a single adenosine: GGGAGGG (or GGGAG_A_GG with a bulge in the second G-tract as in A5U^AD^) (Table 1).

### 2.5. ZmNDPK1 Binds Intermolecular and Intramolecular G4s

ZmNDPK1 binds to G4s that are annealed in Li^+^ with 40-fold weaker affinity [33], but promotes G4 folding upon binding (Figure 9A). To further explore the binding properties of ZmNDPK1 with G-rich DNA that is not pre-formed into an intramolecular G4 conformation, we tested whether or not ZmNDPK1 could bring together two separate DNA strands. When equimolar amounts of 5′ fluorescein (FAM)-labeled *hex4*_A5U and 3′ TAMRA-labeled *hex4*_A5U oligonucleotides were mixed and then annealed either in K^+^ or Li^+^, neither sample exhibited Förster resonance energy transfer (FRET) in the absence of protein (Figure 9B). When ZmNDPK1 was added to the K^+^-annealed oligonucleotide there was no change to the FRET signal. In contrast, the single-labeled oligonucleotides pre-annealed in Li^+^ exhibited FRET when mixed with ZmNDPK1 (Figure 9B). Interestingly, when A5U^R20^ oligonucleotide (20 random bases ending with the 3′ GGGAGGG hairpin) was used instead of *hex4*_A5U, no FRET was observed despite A5U^R20′^s signature of parallel G4 in CD experiments

### 2.6. ZmNDPK1 and Trim_A5U form a Heterogeneous Protein: Nucleic Acid Complex

To gain insight into the mechanism of complex formation between ZmNDPK1 and trim_A5U, we used electron microscopy to visualize the protein alone and in the presence of the G4 oligonucleotide (Figure 10). For ZmNDPK1 alone, we saw uniformly distributed globular protein molecules of the expected size (Figure 10A). After ZmNDPK1 was incubated with trim_A5U we saw the formation of filamentous structures of uniform thickness but various lengths and shapes (Figure 10B). ZmNDPK–trim_A5U complex was then plunge-frozen and images were collected in vitrified ice under the cryogenic conditions (Figure 10C). We saw that filaments were well preserved in ice and uniformly distributed. 2D classification of the particles picked from cryogenic electron microscopy (cryo-EM) images confirmed that the complex had a distinct structure over short distances but was too heterogeneous for further 3D analysis (Figure 10D).

## 3. Discussion

### 3.1. G4s in the Stress Response

G4s are now recognized as important elements in the regulation of intracellular processes related to replication, transcription, translation, splicing, and telomere maintenance [50]. In fact, G4 formation in the promoter of a gene can either inhibit [51,52] or facilitate its transcription [6,53]. In vivo, G4s exist in the context of the double-stranded genome and are regulated through interaction with G4-binding proteins like XPB and telomere end-binding proteins [50,54,55,56]. In vitro, G4 formation is largely driven by the presence of K^+^ or Na^+^, so in addition to possible coordination by proteins, G4 formation is also sensitive to the ionic environment of the cell. In this study, we investigated the properties of the *hex4*_A5U oligonucleotide derived from the G-rich sequence located on the template strand in the 5′ untranslated region of the maize *hexokinase4* gene. This gene is particularly interesting because it has three putative G4s—two on the template strand and one on the coding strand of DNA [16].

### 3.2. Limitations in G4 Characterization Affect Analysis of hex4_A5U

UV-Vis spectrophotometry, CD spectrophotometry, and DMS footprinting are commonly used techniques to verify G4 formation by a given nucleotide sequence. Each has unique strengths, but none is able to unambiguously assess the G4 conformation, and so they must be used together to understand the possible G4 variations of even the simplest G-rich sequence. TDS is a qualitative technique based on UV-Vis spectrophotometry that relies on the hyperchromicity of G4s at 295 nm, but the signal changes qualitatively with the base composition of the nucleic acid [57]. *hex4*_A5U shows a distinct G4 TDS profile only in the presence of K^+^ ions (Figure 1B), whereas the TDS profile of *hex4*_A5U in Na^+^, Li^+^, and Cs^+^ is intermediate between K^+^ and the absence of cations, suggesting formation of a weak G4. Aside from TDS, UV-Vis spectrophotometry can be used to monitor the stability of the G4 by measuring the change in absorbance at 295 nm with increasing temperature [6]. In 100 mM K^+^, *hex4*_A5U had a T_1/2_ of 58 C, showing that it is stable at physiologically relevant temperatures (Figure 1C). We also observed an initial decrease in absorbance at 295 nm for all other cations, suggesting melting of a transient G4, but not in the absence of cations. Overall, these observations show that *hex4*_A5U, expectedly, forms a G4 only if stabilized by a cation, where K^+^ >> Na^+^ > Li^+^ > Cs^+^.

CD spectrophotometry is commonly used to assess the properties of oligonucleotides to give clues about secondary structure. *hex4*_A5U has CD spectra characteristic of a parallel G4 conformation [46] in K^+^, Na^+^, Li^+^, and Cs^+^ (Figure 1D). In the absence of any small cation, CD spectra indicate that the oligonucleotide is disordered (Figure 1D). Interestingly, CD thermal denaturation experiments show that G4s in K^+^ melt as a single species, whereas in Na^+^, Li^+^, and Cs^+^ there is a structural transition evidenced by the shift of the peak maxima from 262 to 255 nm (Figure 3). After the transition, melting profiles for Na^+^, Li^+^, and Cs^+^ resemble that of the oligonucleotides annealed in the absence of cations. The temperature at which this transition occurs is cation-dependent and matches the G4 stability order for cations determined in UV-Vis thermal denaturation experiments: K^+^ >> Na^+^ > Li^+^ > Cs^+^.

Lastly, DMS footprinting provides an additional insight into G4 topology by analyzing solvent-accessible Gs. DMS footprinting showed strong protection of Gs only in K^+^ (Figure 4A, lane 1), whereas protection in Na^+^, Li^+^, and Cs^+^ was limited to the GGGAGGG hairpin at the 3′ end of the sequence (Figure 4A, lanes 2–4). All Gs were completely unprotected in absence of cations, representing a fully unfolded state. We attribute the partial protection in Na^+^, Li^+^, and Cs^+^ to the formation of a weak intermolecular G4 that forms by cation stabilization of the 3′ GGGAGGG hairpins from two DNA molecules. Indeed, the A5U^R20^ oligonucleotide, which has 20 random non-G bases followed by a GGGAGGG hairpin on the 3′ end, does not have a characteristic G4 TDS spectrum (Figure 5B), but does have a parallel G4 CD spectrum in the presence of cations (Figure 5C). We conclude that *hex4*_A5U forms an intramolecular parallel G4 only in the presence of K^+^, whereas it forms a weak intermolecular G4 in the presence of Na^+^, Li^+^, or Cs^+^

### 3.3. hex4_A5U and its Truncated Variant trim_A5U are Highly Polymorphic G4-Forming Sequences

G4 polymorphism is a common, complicating predicting their structures based on sequence alone. Examples of polymorphism include extra G-tracts that can act as a “spare tire” [58]; formation of an ensemble of structures with different topologies [59,60]; variation in number of strands (one, two, or four) and tetrads (two or more); presence of bulges [35]; and loops longer than seven nucleotides [60]. *hex4*_A5U was initially predicted to form from four uninterrupted G-tracts of three sequential Gs (Figure 6A). Instead, DMS footprinting revealed that only two G-tracts were fully protected, whereas G-tract II was not protected and G-tract III was only partially protected (Figure 4A, lane 1). Further, G-tract II was not strictly required for G4 formation in K^+^ (Figure 4A, Figure 5A). To explain this mismatch, we hypothesize that adjacent Gs can be substituted into G-tract II, forming a series of structures with bulged G-tracts that co-exist in solution (Figure 6A). Such a polymorphic system combines G-register exchange [42] with the formation of bulged variants and leads to the apparent absence of protection in G-tract II and only partial protection in G-tract III over the course of DMS labeling. DMS footprinting of trim_A5U revealed that, in this truncated construct, the strong protection of guanines was only for tracts III and IV (Figure 6B, lane 2). The only sequence variant in which we observed complete protection of all 12 guanines involved in G4 core formation was in DMS footprinting of A5U^AH^ construct (Figure 6B, lane 4), in which all extra guanines were substituted by thymidines. In this case, the locked variants described only one possibility of the variations that the native DNA sequence might adopt. We interpreted the data measured on the locked variants to inform us about the ensemble of structures that can possibly form by the native sequence, but it could also be that no single mutation exactly mimics the behavior of the oligonucleotide with the full-length, native *hex4*_A5U sequence

An expanded definition of G4-forming sequences emerges that allows G-tracts to be interrupted by a one-base bulge connected into a continuous region that we call a “G-slide” (Figure 6A). This guanine-rich region can also be mathematically described as a 10-choose-6 combinatorics problem that results in 260 combinations, of which we explored only 13 variants by limiting ourselves to single-bulge interruptions of G-tracts. From our formulation, trim_A5U can form at least 13 different conformers, isolated structurally by point mutagenesis (Figure 6B, Table 1). By all measurements, each resulting variant behaves in a sequence-specific manner that is ultimately predictive of its fold and determines its interaction with the G4-binding protein ZmNDPK1 (Figure 7 and Figure 8). The CD spectrum of the trim_A5U sequence has a minor contribution to the anti-parallel signal when compared to the locked variants (Figure 7B and Figure 8B). This suggests that predominantly antiparallel hybrids (A5U^AE^, A5U^AF^, A5U^AG^, A5U^BF^, and A5U^BG^) as well as unstable variants (A5U^AD^, A5U^CF^, and A5U^CG^) constitute a small fraction of solution conformations. Therefore, the co-existence of parallel G4s with variable G-slide picks (A5U^AH^, A5U^BH^, A5U^CH^, A5U^DH^, and A5U^EH^) represent the majority of conformational states of trim_A5U. Overall, the wild-type conformation is likely determined by the relative stability of the fold and the presence of a GGGAGGG hairpin that favors the formation of a parallel G4 (Figure 7B, Table 1) [61].

### 3.4. The G4-Binding Protein ZmNDPK1 Recognizes a Subset of Conformations Adopted by hex4_A5U DNA and Forms Filamentous Structures upon Binding

DNA is associated with protein binding partners within the nucleus. ZmNDPK1, a plant homolog of human NM23-H2, interacts with *hex4*_A5U with high affinity and specificity [33]. Despite the analogy between plant and human NDPKs binding to G-rich DNA sequences [62,63,64], we do not know how they interact or, until now, what structural motifs direct binding. ZmNDPK1 does not have a single preferred G4 conformation, but binds more specifically to parallel G4s that contain the GGGAGGG motif with or without bulges (Figure 8, Table 1). Additionally, ZmNDPK1 recognizes the structural element that gives rise to weak G4 signals in sub-optimal G4-promoting ions (i.e., Li^+^), perhaps a transitory guanine hairpin [65], and then facilitates bimolecular G4 formation (Figure 9A,B).

Electron microscopy of the ZmNDPK1–trim_A5U complex revealed its assembly into filamentous structures (Figure 10B,C). These structures differed in their lengths, but not thickness. 2D classification of particles from cryoEM images provided a low-resolution look into organization of this complex (Figure 10D). We can see that the complex is highly flexible, and poorly resolved, which made it not possible to distinguish between protein and DNA densities in our 2D classes. One thing was clear—the complex of Z4sNDPK1–trim_A5U was not as simple as two G4s per one hexamer as we predicted from the stoichiometry determined biochemically in solution.

### 3.5. Generalization of G4 Heterogeneity across Domains of Life

The ability for G4 DNA to form multiple conformations with protein-binding specificity is not unique to maize, so characterizing the range of morphologies that long, non-continuous G-rich stretches can adopt is relevant in possibly exploiting the phenomenon for anti-microbial or anti-viral therapies. For example, a common bacterial (G_4_CT)_3_G_4_ motif associated with antigenic variation exhibits cation-stabilizing, concentration-dependent conformational variability that is sequence dependent [66]. The striking similarity to the phenomenon we observed with the *hex4*_A5U motif suggests that this conformational variability may well influence how the sequence interacts with its protein partners in microorganisms. This idea is supported by the observation that in *Neisseria gonorrhoeae*, a monomeric—but not dimeric—parallel, G4 binds RecA to direct recombination at the pilin expression locus during antigenic switching [67]. Further, the ability for a G4 linked to nitrate assimilation in *Paracoccus denitrificans* to form inter- or intramolecular G4s (i.e., G4′s insensitivity to NH_4_^+^) and a mix of parallel and anti-parallel conformations in solution suggests plasticity could also play a role in this microorganism [68]. This feature is not limited to microorganisms—the G-rich proviral HIV-1 U3 DNA forms polymorphic G4 structures that have different Sp1-binding capabilities that are proposed to fine-tune transcription [69]. Human G4s including c-myc [38], RET [70,71,72], VEGF [73], and BCL-2 [74] also have the ability to form multiple conformations. Indeed, a minimal version composed of four Gs in a single G-tract where the 5′ or 3′ G can swap into the three-G stretch of the slide region has been described as a slippage of the G-tract in *c-myc* [38]. Similarly, a specific instance of the slide can be seen in the oxidative protection mechanism described as the spare tire, where a fifth terminal G-tract can slide into place, positioning the fourth G-tract in a long loop that allows repair of oxidatively damaged Gs [58]. Here, we have generalized these specific examples into a model that allows Gs from long, non-continuous G stretches to slide into the G4 stack, creating a range of G4 conformations that have unique properties and specific responses to a G4-binding protein (Figure 11). Such heterogeneity in G4 formation is an innate biophysical property of G4s that is likely conserved from prokaryotes to eukaryotes.

## 4. Materials and Methods

### 4.1. Oligonucleotide and Protein Preparation

All oligonucleotides were purchased from Eurofins MWG Operon LLC (Huntsville, AL, USA) as salt-free (non-labeled oligonucleotides) or HPLC-purified (fluorescently labeled oligonucleotides) and used without further purification. Base positions in oligonucleotide variants were numbered according to the positions in the *hex4*_A5U sequence [33]. Unless indicated otherwise, oligonucleotides were annealed by heating to 95 °C, then slowly cooled overnight to room temperature in 10 mM tetrabutyl ammonium phosphate (TBA, pH 7.5) buffer with or without 100 mM salt (KCl, LiCl, CsCl, or NaCl), or in water alone. Recombinant ZmNDPK1 protein was purified as previously described [33].

### 4.2. Absorption Spectrophotometry

Non-labeled oligonucleotides were annealed at 10 μM concentration and diluted to 2.5 μM before data collection. All UV-Vis experiments were performed on a Cary 300 Bio UV/Vis spectrophotometer equipped with a Peltier temperature controller (Agilent Technology, Santa Clara, CA, USA). For thermal difference spectroscopy (TDS), a first spectrum was collected at 25 °C, samples were heated to 95 °C, and a second spectrum was collected. TDS was calculated by subtracting the 25 °C spectrum from the 95 °C spectrum and normalizing the maximum peak to an absorbance of 1 and the absorbance at 330 nm to 0. For thermal denaturation experiments, the absorbance at 295 nm was monitored in the temperature range from 25 to 95 °C at a heating rate of 0.5 °C/min. Data were normalized to a maximum of 1.

### 4.3. Circular Dichroism Spectrophotometry

Non-labeled oligonucleotides were annealed at 10 μM concentration and used without further dilution. Circular dichroism (CD) spectra were collected on an Aviv 202 CD spectrometer (Aviv Biomedical, Lakewood, NJ, USA). Single temperature experiments were performed at 25 °C over a 200–330 nm range with a 3-s average time. The same parameters were used for thermal denaturation experiments in which measurements were made between 10 and 95 °C with a 5 °C increment between measurements after a 10-min equilibration. All spectra were background corrected against blank buffer and normalized to have zero ellipticity at 330 nm.

### 4.4. Dimethyl Sulfate (DMS) Footprinting

*O*ligonucleotides with a 5′ 6-carboxyfluorescein (FAM) modification were annealed at 10 μM concentration and diluted to 500 nM concentration prior to DMS treatment. Samples were treated with 1% DMS for 5 min at 25 °C and stopped by adding 25 μL of quench solution (1.5 M sodium acetate pH 7.0, 1 M β-mercaptoethanol and 100 μg/mL calf thymus DNA). DNA was ethanol-precipitated and pellets were resuspended in 100 μL of 1 M piperidine, incubated for 15 min at 95 °C, and dried in a rotary centrifuge. Dried samples were washed with distilled water, resuspended in alkaline sequencing dye (80% formamide, 10 mM NaOH, 0.005% bromophenol blue), and heated to 95 °C for 3 min. Cleavage products were resolved on a 17.5% polyacrylamide denaturing gel (4 M urea, 0.5x tris-borate-EDTA, 0.4 mm thick, 33 × 39 cm, 29:1 acrylamide/bisacrylamide) run for 1.5 h at a constant 50 W power. Glass plates were separated and the gel was imaged on a GE Typhoon scanner (GE Healthcare Bio-Sciences, Pittsburg, PA, USA) in fluorescence mode using a 488-nm excitation wavelength and a 520-nm band pass filter.

### 4.5. Nitrocellulose Filter Binding Assays for ZmNDPK1/G4 DNA Binding Affinity Analysis

For binding-affinity determination, we used a modified slot-blot binding assay as previously described [33], substituting a 5′ biotin label with a 5′ carboxyfluorescein. We used the same approach to determine the efficiency of ZmNDPK1 binding to labeled oligonucleotides in the presence of competitor oligonucleotides. All oligonucleotides were annealed in 10 mM TBA (pH 7.5) + 20 mM KCl. Labeled oligonucleotide at 1 nM was mixed with 100 nM competitor oligonucleotide and 5 nM ZmNDPK1. Reactions were incubated for 60 min and applied to the slot-blot apparatus, where the solution first passes through a negatively charged nitrocellulose membrane (Hybond-C Exatra 0.45 μM pore size, GE Healthcare Life Sciences, Piscataway, NJ, USA) that retains protein and protein–DNA complex. Unbound DNA was then captured by a positively charged nylon membrane (Nytran N 0.45 μM pore size, GE Healthcare Life Sciences, Piscataway, NJ, USA). Membranes were dried and scanned on a GE Typhoon scanner in fluorescence mode using a 488-nm excitation wavelength and a 520-nm band pass filter. Images were background corrected and the intensities of the bands were determined in ImageJ. Competition efficiency was calculated from the retention percentage of the fluorescent probe on nitrocellulose against zero competitor control.

### 4.6. Analytical Ultracentrifugation (AUC)

Sedimentation experiments were carried out in a Beckman Coulter ProteomeLab XL-1 analytical ultracentrifuge using an AN60-Ti rotor and double-sector quartz cells. We loaded 420 μL of annealed oligonucleotides at 1 μm into sample sectors and 430 μL of corresponding annealing buffers into reference sectors. Initial scans and rotor calibrations were performed at 3000 rpm and a 260-nm wavelength. Data were collected at 58,000 rpm and analyzed using Ultrascan III software [48].

### 4.7. Fluorescence Resonance Energy Transfer (FRET)

*hex4*_A5U oligonucleotides were labeled with either 5′ 6-carboxyfluorescein (*hex4*_A5U-5F) or 3′ carboxytetramethylrhodamine (*hex4*_A5U-3T) or both fluorophores (*hex4*_A5U-5F3T). Reactions were set up in triplicate in 96-well Nunclon plates (Thermo Fisher Scientific, Waltham, MA) containing 200 nM of either *hex4*_A5U-5F3T or a mix of 100 nM *hex4*_A5U-5F + 100 nM *hex4*_A5U-3T annealed in 10 mM TBA (pH 7.5) + 100 mM KCl or 100 mM LiCl. Protein was added at 0, 200 nM, 500 nM, or 1000 nM concentrations and incubated for 1 h at 4 °C before data collection. Data were collected on a Spectramax M5^e^ Multi-Mode Microplate Reader (Molecular Devices, Sunnyvale, CA, USA) and processing was performed as previously described [33]. Labeling and data collection for A5U^R20^ oligonucleotides were done as described for *hex4*_A5U.

### 4.8. Electron Microscopy (EM)

NDPK–G4 complex was assembled by mixing 3 µM ZmNDPK1 and 6 µM *hex4*_A5U in a buffer containing 10 mM Hepes pH 7.5 and 50 mM KCl. For negative staining, the mixture was applied to plasma-cleaned CF200-Cu carbon-coated copper grids (Electron Microscopy Sciences, Hatfield, PA, USA), incubated for 60 s, washed 3x with distilled water, and stained for 60 s with 1% uranyl-acetate. Images were collected on a FEI/Philips CM120 Biotwin electron microscope (Thermo Fisher Scientific, Waltham, MA, USA) at 40,000 magnification (2.8 Å/px). For cryo-electron microscopy (cryoEM) the mixture was applied to the carbon side of the plasma-cleaned Quantifoil R2/2 grids (Electron Microscopy Sciences, Hatfield, PA, USA) and plunged into liquid ethane using FEI Vitrobot (Thermo Fisher Scientific, Waltham, MA, USA). Plunge-frozen grids were imaged on an FEI Titan Krios (Thermo Fisher Scientific, Waltham, MA, USA) equipped with a DE20 direct electron detector camera (Direct Electron, San Diego, CA, USA) at 37,000 magnification and 0.99 Å pixel size. Automatic data acquisition was set up using Leginon software [75]. Images were collected with a 1.5–3.5 µm defocus range. Particles were manually picked from the images using a Leginon particle picker. Particle coordinates were used to create a particle stack of ~30.000 particles. Particle stack was 2D classified in cryoSPARC [76] into 30 classes.

## Figures and Tables

**Figure 1 molecules-24-01988-f001:**
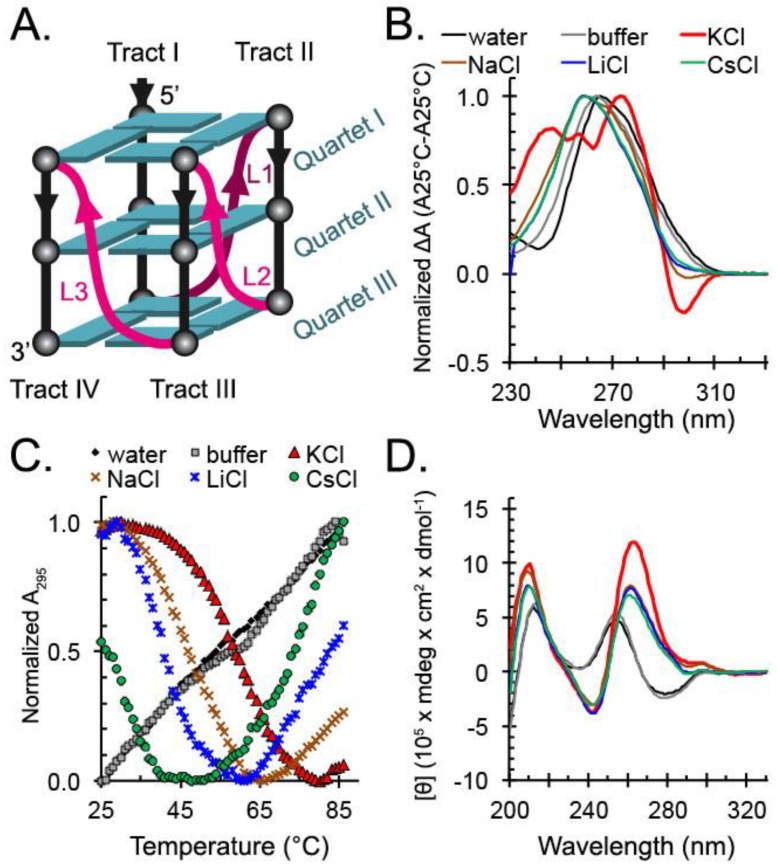
Spectroscopic analysis of a G-quadruplex (G4) formation by *hex4*_A5U. Oligonucleotides annealed in water (black), 10 mM tetrabutyl ammonium phosphate (TBA) buffer pH 7.5 (gray), or 10 mM TBA buffer supplemented with 100 mM of KCl (red), NaCl (brown), LiCl (blue), or CsCl (green). (**A**) Schematic representation of a canonical parallel unimolecular G4. Four tracts of three consecutive guanines (spheres) form three stacked G-quartets (cyan) stabilized by a monovalent cation. L1, L2, and L3 are lateral loops (magenta). (**B**) Normalized thermal difference spectra show formation of G4s, indicated by a negative peak at 295 nm. A prominent negative peak is observed only in KCl, and is absent in water or buffer, with intermediate values observed for NaCl, LiCl, and CsCl. (**C**) Thermal melting measured the cation-dependent stability of the G4 structures. G4s formed in KCl were the most stable, with a T_1/2_ of 58 °C, followed by 50 °C for NaCl, 42 °C for LiCl, and <30 °C for CsCl. A linear increase in absorbance at 295 nm for oligonucleotides annealed in the absence of cation indicates that no G4 structures formed. (**D**) Circular Dichroism (CD) spectra show the formation of parallel G4s in the presence of cations. Peak maxima at 262 nm and minima at 242 nm were the hallmarks of the parallel G4s and were observed in KCl, NaCl, LiCl, and CsCl.

**Figure 2 molecules-24-01988-f002:**
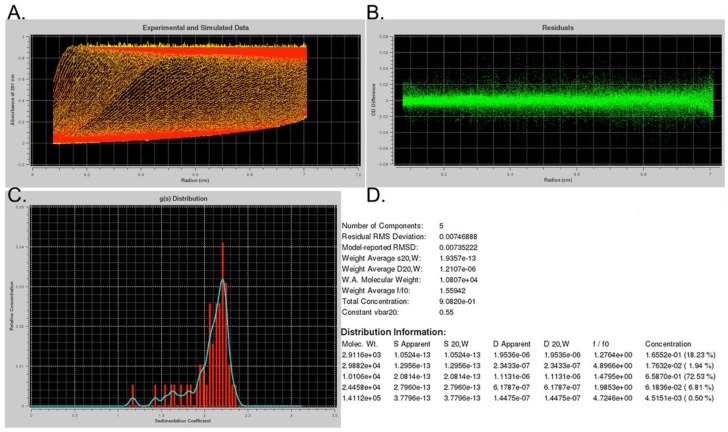
Analytical ultracentrifugation (AUC) of *hex4*_A5U annealed in KCl shows formation of a compact structure. (**A**) Raw sedimentation scans (yellow) overlaid with the calculated fit (red). (**B**) Residuals between the fit and the model plot showing their random nature. (**C**) Relative concentration and distribution of the species with different sedimentation coefficients. (**D**) Summary table of the contents of the solution after genetic algorithm analysis as implemented in Ultrascan 3 [47,48].

**Figure 3 molecules-24-01988-f003:**
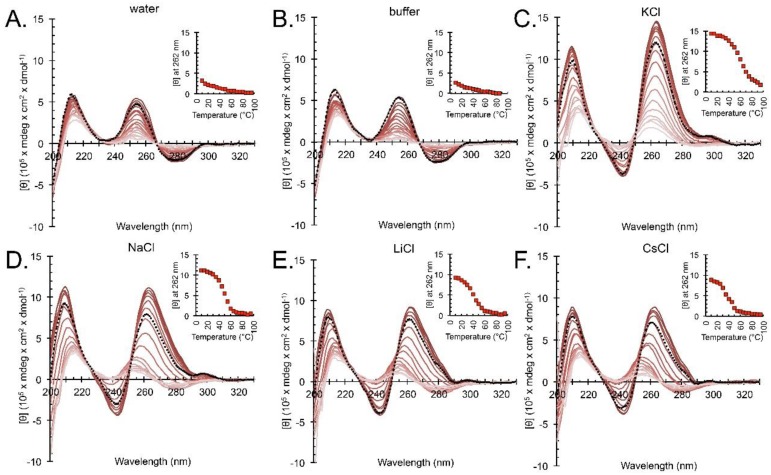
CD thermal denaturation shows reversible structural transition of a G4 formed by *hex4*_A5U with different cations. Oligonucleotides were annealed in 10 mM TBA buffer supplemented with 100 mM of KCl (**A**), NaCl (**B**), LiCl (**C**), CsCl (**D**), buffer alone (**E**), or water (**F**). In all conditions, we observed an overall decrease in CD signal intensity with an increase in temperature. Melting G4s annealed in KCl resulted in a sigmoidal curve with a T_1/2_ of 58 °C. Melting in NaCl, LiCl, and CsCl revealed a two-state behavior indicative of a structural transition from a G4 to ssDNA in which a sigmoidal phase was followed by a linear phase. Melts for water and TBA buffer alone were linear and represented unstacking of the ssDNA bases. Thermal unfolding of the secondary structure was reversible, indicated by the dashed black line that corresponds to spectra collected immediately after the samples were cooled to 20 °C. Insets: plots of molar ellipticity at 262 nm versus temperature.

**Figure 4 molecules-24-01988-f004:**
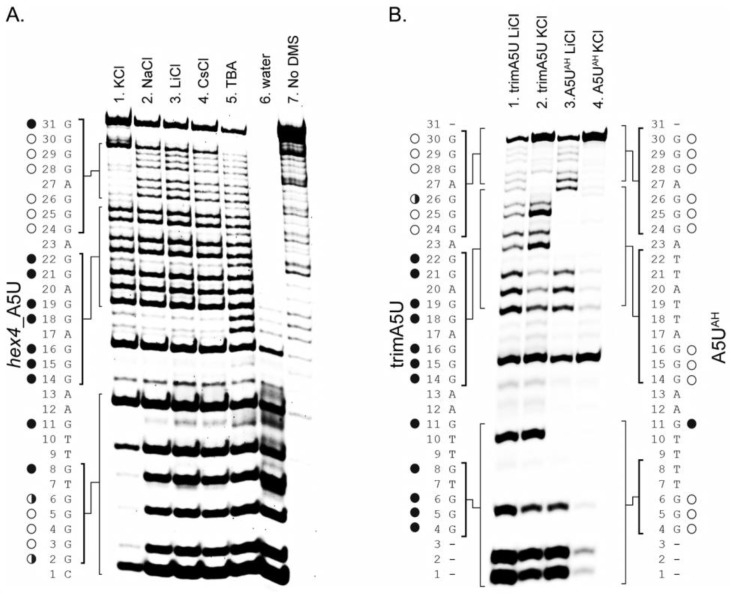
Dimethyl sulfate (DMS) footprinting of *hex4*_A5U and its variations reveal guanines involved in G4 core formation. (**A**) Missing bands on a gel indicate guanines protected from DMS labeling. A distinct footprinting pattern is observed only for the KCl sample (lane 1). In NaCl, LiCl, and CsCl, partial protection is observed for the GGGAGGG hairpin at the 5′ end of the oligonucleotide (lanes 2–4). In TBA, all guanines are digested evenly and in water alone the sample is overdigested. Circles (left) indicate guanines that are protected (○), partially protected (◗), or overdigested (●) when treated in KCl. (**B**) *hex4*_A5U was trimmed by removing bases 1, 2, 3, and 31, resulting in trimA5U construct. trimA5U was further altered by substituting G_8_, G_18_, G_19_, G_21_, and G_22_ with thymidines, resulting in A5U^AH^ construct. Both trim_A5U and A5U^AH^ oligonucleotides were subjected to DMS footprinting in KCl or LiCl.

**Figure 5 molecules-24-01988-f005:**
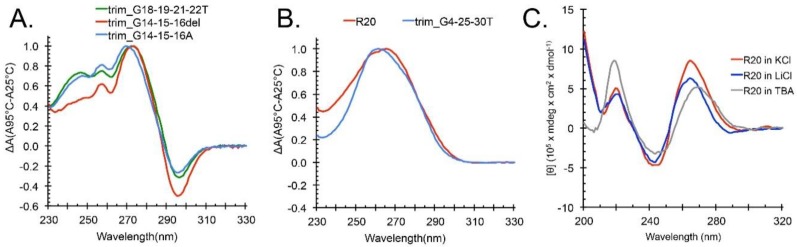
Preliminary mutagenesis of the trim_A5U oligonucleotide. Oligonucleotides were annealed in a buffer containing 10 mM TBA pH 7.5 supplemented with 100 mM KCl or LiCl. (**A**) trim_A5U variants with the deletion or substitution of tract II guanines with adenines form G4s. (**B**) trim_G4-25-30T oligonucleotide with point mutations in tracts I, III, and IV and A5U^R20^ oligonucleotide with a randomized sequence upstream of G-tract II did not form stable G4s. (**C**) G4-characteristic CD spectrum is observed only when A5U^R20^ oligonucleotides are annealed in the presence of cations, but not in the TBA buffer alone.

**Figure 6 molecules-24-01988-f006:**
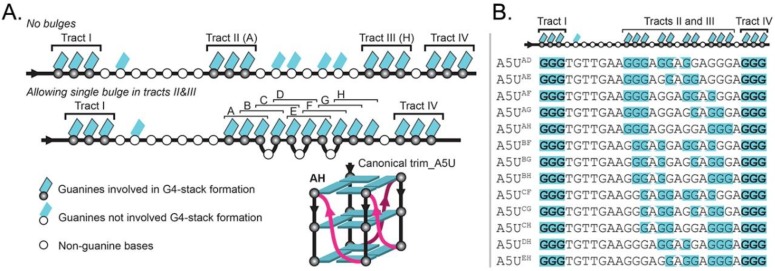
Extended model of G4 formation by trim_A5U allowing one bulge in G-tract. (**A**) Comparison between the canonical model and the extended G4 model. The extended model allows longer loops and a one-base-bulge interruption of G-tracts. Under the canonical model there is only one possible fold that can be adopted by trim_A5U to form a G4 core using tracts that do not contain bulges (i.e., tracts II (labeled A) and III (labeled H)). Under the extended model, trim_A5U can potentially form 13 different G4 core folds (including a canonical fold A5U^AH^) with fixed tracts I and IV and the potential of one-base bulges in tracts II and III. (**B**) Guanines that can be involved in formation of the G4 core in the extended model are highlighted.

**Figure 7 molecules-24-01988-f007:**
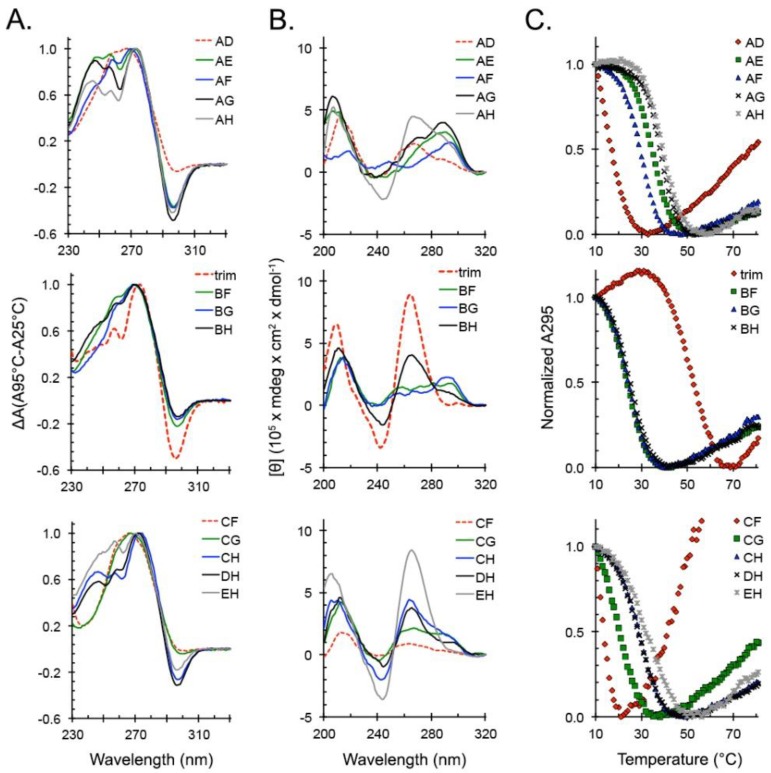
Spectroscopic analysis of a G-quadruplex (G4) formation by trim_A5U-locked variants. **t**rim_A5U variants based on the extended model (derived from Figure 6B) were tested for their ability to form G4s. Guanines were substituted with thymidines to preclude their involvement in G4 core formation. (**A**) With the exception of A5U^AD^, A5U^CF^, and A5U^CG^, all trim_A5U variants have a prominent negative peak at 295 nm, indicating G4 formation. (**B**) Thermal meltings monitored at 295 nm show that all locked variants formed G4s with different stabilities; however, A5U^AD^, A5U^BF^, A5U^BG^, A5U^BH^, A5U^CF^, A5U^CG^, formed weak G4s with T_1/2_ < 30 °C. (**C**) CD spectra show that locked variants formed G4s with different topologies. Variants A5U^AD^, A5U^AH^, A5U^BH^, A5U^CF^, A5U^CH^, A5U^DH^, and A5U^EH^ formed parallel G4s with a major peak at 262 nm. A5U^AE^, A5U^AF^, A5U^AG^, A5U^BF^, and A5U^BG^ formed antiparallel hybrid G4s with a major peak at 292 nm. A5U^CG^ has a mixed spectra with similar ellipticity at 262 and 292 nm.

**Figure 8 molecules-24-01988-f008:**
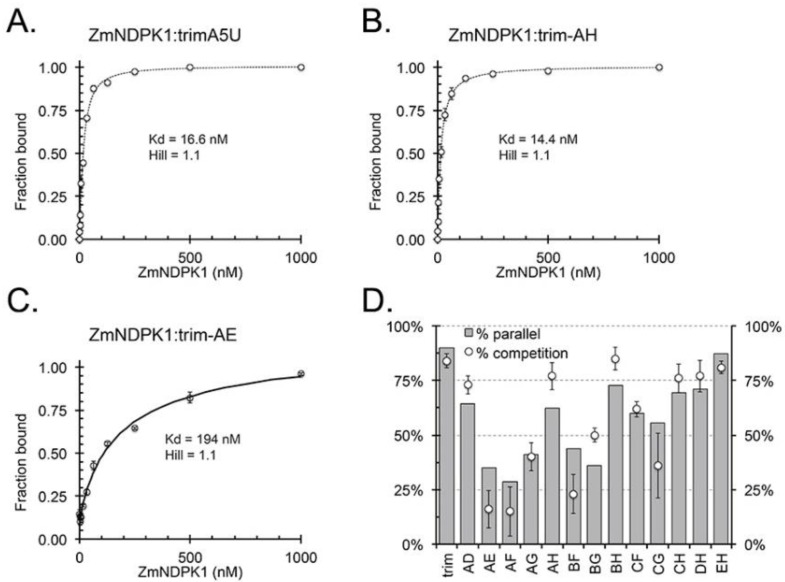
G4-binding protein ZmNDPK1 preferentially binds to the parallel locked variants of trim_A5U. (**A**) Binding of ZmNDPK1 to the trim_A5U caused the retention of the fluorescently labeled oligonucleotide on the nitrocellulose. (**B**) Binding of ZmNDPK1 to the trim_AH caused the retention of the fluorescently labeled oligonucleotide on the nitrocellulose. (**C**) Binding of ZmNDPK1 to the trim_AE caused the retention of the fluorescently labeled oligonucleotide on the nitrocellulose. (**D**) Competition efficiency was calculated as the amount of the probe retained on the nitrocellulose compared to the no-competitor control. Only parallel G4-locked variants competed with high efficiency: A5U^AD^, A5U^AH^, A5U^BH^, A5U^CF^, A5U^CH^, A5U^DH^, and A5U^EH^.

**Figure 9 molecules-24-01988-f009:**
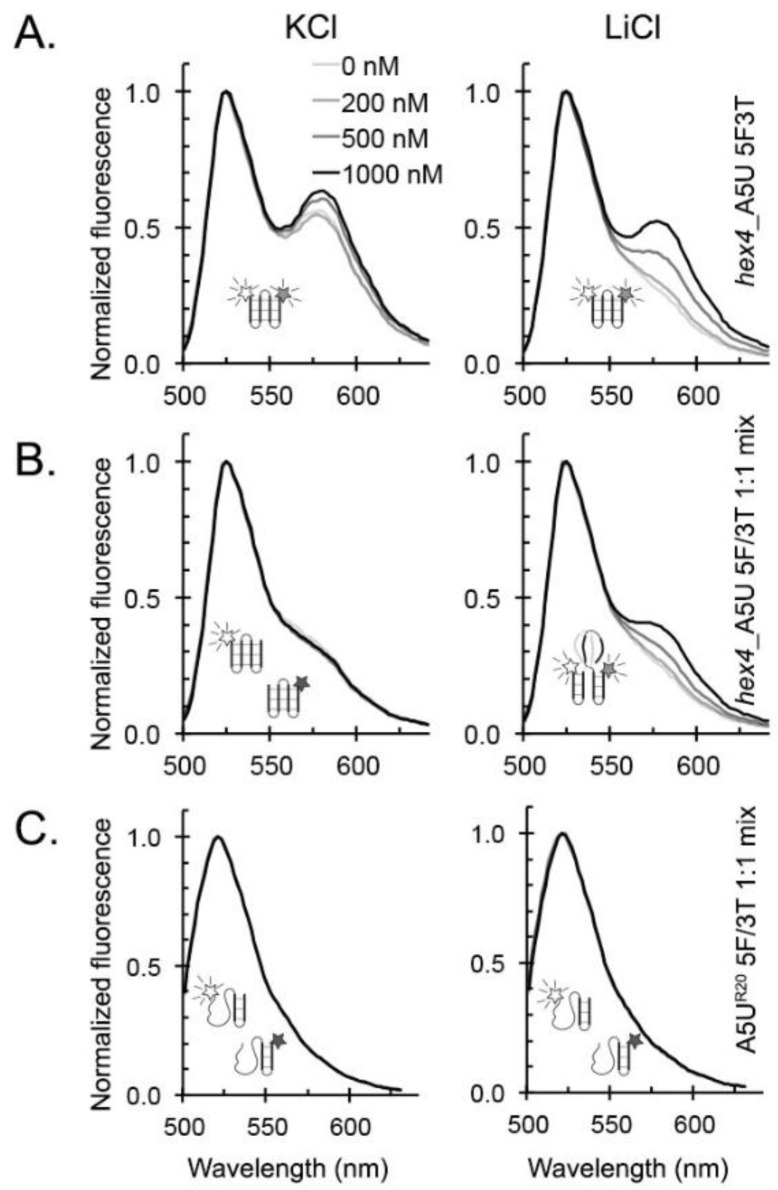
ZmNDPK1 binds to intermolecular and intramolecular G4s. Fluorescence emission data were collected by exciting the FAM fluorophore, and resulting plots were normalized to the peak maxima of 1 to better visualize the changes. (**A**) *hex4*_A5U_5F3T: dual-labeled oligonucleotides. When annealed in KCl, the FRET signal changed little with increasing protein concentration. When annealed in LiCl, the FRET signal increased with increasing protein concentration. (**B**) *hex4*_A5U_5F/3T: two single-labeled oligonucleotides. When annealed in KCl, FRET did not change with added protein. When annealed in LiCl, the FRET signal increased with increasing in protein concentration. (**C**) A5UR20 5F/3T: two single-labeled oligonucleotides, with 20 random non-G bases ending with a GGGAGGG hairpin. When annealed in either KCl or LiCl, the FRET signal did not change with added protein.

**Figure 10 molecules-24-01988-f010:**
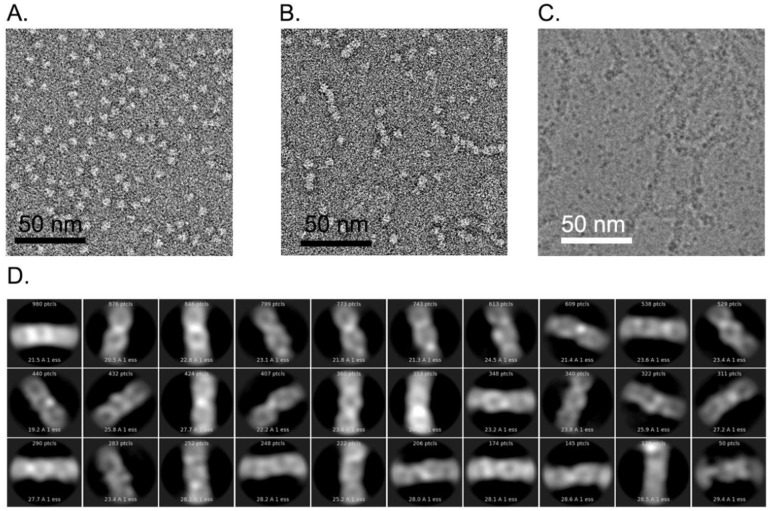
Electron microscopy of the complex between ZmNDPK1 and trim_A5U. (**A**) Image of a negatively stained ZmNDPK1 alone. (**B**) Image of the negatively stained ZmNDPK1 in complex with trim_A5U. (**C**) CryoEM image of ZmNDPK1 in complex with trim_A5U stain in vitreous ice. (**D**) Results of 2D classification of 30.000 filament segments picked from the cryoEM images.

**Figure 11 molecules-24-01988-f011:**
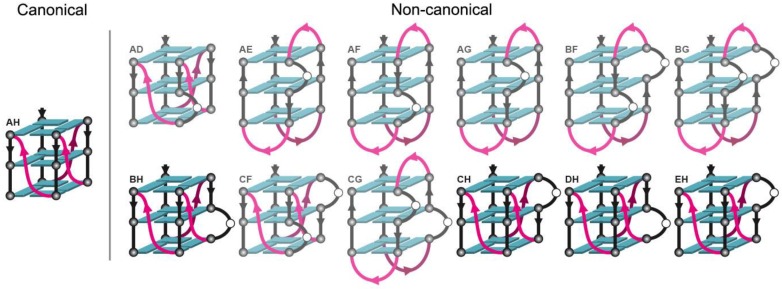
Possible topologies that can be adopted by the trim_A5U oligonucleotides. Out of 13 conformation possibilities predicted by the extended model, only one is canonical (A5U^A^), while 12 others contain a bulge in G-tract II, III, or in both G-tracts. ZmNDPK1 binds to the variants with the conserved GGGAGGG hairpin (contrasted models).

**Table 1 molecules-24-01988-t001:** Summary of the properties of trim_A5U-locked variants. Gs that participate in G4 formation are in bold and G-tracts are underlined and bold. Mutated residues are in lowercase.

Oligonucleotide	Sequence	CD	*Tm*	Competition %
A5U^AD^	**GGG**TtTTGAA**GGG**A**GG**A**G**tAtttA**GGG**	parallel	<30	73
A5U^AE^	**GGG**TtTTGAA**GGG**At**G**A**GG**AtttA**GGG**	anti-h	37	16
A5U^AF^	**GGG**TtTTGAA**GGG**AttA**GG**A**G**ttA**GGG**	anti-h	~30	15
A5U^AG^	**GGG**TtTTGAA**GGG**AttAt**G**A**GG**tA**GGG**	anti-h	40	40
A5U^AH^ (canonical)	**GGG**TtTTGAA**GGG**AttAttA**GGG**A**GGG**	parallel	42	77
A5U^BF^	**GGG**TtTTGAAt**GG**A**G**tA**GG**A**G**ttA**GGG**	anti-h	<30	23
A5U^BG^	**GGG**TtTTGAAt**GG**A**G**tAt**G**A**GG**tA**GGG**	anti-h	<30	50
A5U^BH^	**GGG**TtTTGAAt**GG**A**G**tAttA**GGG**A**GGG**	parallel	<30	85
A5U^CF^	**GGG**TtTTGAAtt**G**A**GG**A**GG**A**G**ttA**GGG**	parallel	<30	62
A5U^CG^	**GGG**TtTTGAAtt**G**A**GG**At**G**A**GG**tA**GGG**	mixed	<30	36
A5U^CH^	**GGG**TtTTGAAtt**G**A**GG**AttA**GGG**A**GGG**	parallel	~30	76
A5U^DH^	**GGG**TtTTGAAtttA**GG**A**G**tA**GGG**A**GGG**	parallel	~30	77
A5U^EH^	**GGG**TtTTGAAtttAt**G**A**GG**A**GGG**A**GGG**	parallel	35	81

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
