# Peer review of "Bulged and Canonical G-Quadruplex Conformations Determine NDPK Binding Specificity"

_molecules, 2019, doi:10.3390/molecules24101988_

Reviewer 1 Report

The manuscript from demonstrated sequence from hexokinase gene could adopt multiple G4 structures. They also investigated the binding reaction of ZmNDPK1 protein with the G4 structure. However, there is no solid evidence to support the authors conclusion. Current data such as CD, TDS and DMS footprint are not sufficient to support the existence of multiple G4 structures including non-canonical bulged G-quadruplex. Also, there is no evidence that the G4 structure formed by the mutant sequences adopted the same structure as native DNA sequences.

p6, line184, “DMS footprinting of trim_A5U construct showed a different footprinting pattern,” indicating G-quadruplex structure is different with hex4_A5U any more. Thus, fig.4B is not meaningful to evaluate the G-quadruplex formed by hex4_A5U sequence.

In fig 9, “ZmNDPK1 binds to and stabilizes intermolecular and intramolecular G4s”, how do authors get the information for stabilization of G-quadruplex.

Some of the authors' explanations are difficult to follow. For example, EM image in fig.10, it is difficult or impossible to discuss conformational heterogeneity of G-quadruplex.

In fig.11, the authors described “canonical” and “non-canonical”, however researchers in G-quadruplex field maybe unable to accept this, because usually “canonical” is for duplex structure, not for G-quadruplex. G-quadruplex structure is “non-canonical” structure.

Author Response

Reviewer 1:

The manuscript from demonstrated sequence from hexokinase gene could adopt multiple G4 structures. They also investigated the binding reaction of ZmNDPK1 protein with the G4 structure. However, there is no solid evidence to support the authors conclusion. Current data such as CD, TDS and DMS footprint are not sufficient to support the existence of multiple G4 structures including non-canonical bulged G-quadruplex. Also, there is no evidence that the G4 structure formed by the mutant sequences adopted the same structure as native DNA sequences. 

We agree that alone, none of these techniques is sufficient to characterize G4 structure. Nevertheless, with something as polymorphic as the native hex4_A5U (Figure 4) and that forms such variable protein:nucleic acid structures (Figure 10), structure determination by X-ray crystallography, cryo-EM, or NMR is not possible. Therefore, we designed this biochemical study to compare the different techniques, which have different strengths and weaknesses, and then to look at the totality of our observations. We agree that the variant G4s do behave differently than the native hex4_A5U because they adopt a single structure, whereas the native DNA is likely a mixture of many populations of structures based on the complex DMS footprint (Figure 4). We have clarified our interpretation in the introduction and discussion to emphasize these points.

Introduction Paragraph 4, lines 65-66 now reads: “Nonetheless, these techniques individually fall short in defining a single state of the oligonucleotide so must be used together to understand the G4 forming potential of any given G-rich stretch of nucleic acid.”

Discussion Paragraph 2, lines 394-397 now reads “UV-Vis spectrophotometry, CD spectrophotometry and DMS footprinting are commonly used techniques to verify G4 formation by a given nucleotide sequence; each has unique strengths but none is able to unambiguously assess the G4 conformation so must be used together to understand the possible G4 variations of even the simplest G-rich sequence.”

Discussion Paragraph 5, lines 447-449 now read “The only sequence variant in which we observe complete protection of all 12 guanines involved in G4 core formation is in DMS footprinting of A5UAHconstruct (Figure 6B, lane 4), where there all extra guanines were substituted by thymidines. In this case, the locked variants describe only one possibility of the variations that the native DNA sequence might adopt.”

・p6, line184, “DMS footprinting of trim_A5U construct showed a different footprinting pattern,” indicating G-quadruplex structure is different with hex4_A5U any more. Thus, fig.4B is not meaningful to evaluate the G-quadruplex formed by hex4_A5U sequence.

The author’s aim in this part of the experiment was to simplify the analysis of the core of the A5U sequence. We modified our sentence to make our motivation clearer, so that phrase (p. 7 line 216) now reads:

“DMS footprinting of trim_A5U construct showed a less complicated footprinting pattern...”

・In fig 9, “ZmNDPK1 binds to and stabilizes intermolecular and intramolecular G4s”, how do authors get the information for stabilization of G-quadruplex.

We show in this experiment that intermolecular G4s readily form with KCl but not LiCl. However, adding ZmNDPK1 allows productive G4 formation. We altered the Figure legend accordingly (page 12 line 349) so it now reads

“Figure 9: ZmNDPK1 binds to intermolecular and intramolecular G4s.” 

・Some of the authors' explanations are difficult to follow. For example, EM image in fig.10, it is difficult or impossible to discuss conformational heterogeneity of G-quadruplex.

Our intention in this experiment was not to explore the polymorphic nature of the G4 itself but, rather, the complex between the G4 and ZmNDPK1. Together, the protein:nucleic acid complex form an incredibly heterogeneous population of filaments that can be aligned and averaged in two dimensions over short stretches but are not sufficiently homogenous for 3D structure determination. We clarified our language to emphasize this point (page 12 lines 359-369).

“2.6 ZmNDPK1 and trim_A5U form a heterogeneous protein:nucleic acid complex.

To gain the insight into the mechanism of complex formation between ZmNDPK1 and trim_A5U we used electron microscopy to visualize the protein alone and in the presence of the G4 oligonucleotide (Figure 10). For ZmNDPK1 alone, we saw uniformly distributed globular protein molecules of the expected size (Figure 10A). After ZmNDPK1 was incubated with trim_A5U we saw formation of filamentous structures of uniform thickness but various lengths and shapes (Figure 10B). ZmNDPK:trim_A5U complex was then plunge-frozen and images were collected in vitrified ice under the cryogenic conditions (Figure 10C). We saw that filaments were well preserved in ice and uniformly distributed. 2D-classification of the particles picked from cryoEM data confirmed that complex has a distinct structure over short distances but was too heterogeneous for further 3D analysis (Figure 10D).”

・In fig.11, the authors described “canonical” and “non-canonical”, however researchers in G-quadruplex field maybe unable to accept this, because usually “canonical” is for duplex structure, not for G-quadruplex. G-quadruplex structure is “non-canonical” structure.

The authors have found numerous publications in the G4 field where “canonical” is used to describe a continuous G4 whereas “non-canonical” is used to describe G4s formed by non-continuous stretches of G’s. Some examples include:

https://elifesciences.org/articles/26884Non-Canonical G-quadruplexes cause the hCEB1 minisatellite instability in Saccharomyces cerevisiae” Piazza, et al 2017

https://academic.oup.com/nar/article/43/20/9575/1395502Directly lighting up RNA G-quadruplexes from test tubes to living human cells” Xu, et al, 2015

https://advances.sciencemag.org/content/4/8/eaat3007Encoding canonical DNA quadruplex structure” Dvorkin, et al, 2018

Reviewer 2 Report

Review attached

Author Response

Reviewer 2:

In this manuscript, Kopylov et al. combined CD and UV spectrophotometry studies with DMS footprinting and point mutations insertion experiments to characterize the G4-forming oligonucleotide hex4_A5U, whose sequence has been extracted from the 5' untranslated region of the maize hexokinase4 gene. The authors also studied the interaction between ZmNDPK1 and the G4-forming oligonucleotides, proving that a subset of the potential hex4_A5U G4 conformers formed complexes with the protein with nanomolar affinity. Notably, ZmNDPK1 proved also to induce G4 formation in some G4-forming oligonucleotides in Li+-rich solutions. Taken together, the obtained results showed for the hex4_A5U sequence an unprecedented level of polymorphism that can be explained in terms of topological isomers and G-register exchange concept, allowing a large number of non-canonical bulged G4 conformations. In the final discussion, the authors hypothesize that the observed polymorphism is a universal property of G4-forming sequences in eukaryotic as well as prokaryotic genomes. 

The work is well written, includes a large number of assays and use of different techniques and all the experiments are clearly and correctly described. In my opinion, it is very interesting and original, adding a lot of new data useful for understanding the structural properties of the selected G4-forming oligonucleotides and get an insight into their functional roles. 

, this work can be published in Molecules after some (minor) revisions. The following points should be addressed before publication. 

1) When in the Introduction the authors say that: “Further, functional roles in regulating transcription and replication continue to be identified from bacteria to mammals”should also mention the emerging role of G-quadruplexes in viruses, including recent works/reviews (e.g., E.Ruggiero, S. N Richter, Nucleic Acids Research, 46(7) 2018, 3270 3283). Also the role of G4-based aptamers in therapeutics and diagnostics should be emphasized, for example mentioning recent articles/reviews in the field. 

To address both points, we expanded Introduction Paragraph 2 (page 1-2 lines37-50)

G4s are identified throughout microorganisms, viral, mammalian, and plant genomes at similar, but not identical, loci. In bacteria, G4s are enriched in regulatory sequences as well as transfer, non-coding and messenger RNAs [12]. In viruses, G4s that are conserved across viral classes are found in gene promoters and long terminal repeats, implicating them in gene expression regulation and viral latency [13,14]. In humans, G4s are enriched just upstreamof transcription start sites (TSSes) as well as in introns near intron-exon boundaries, more commonly found in the sense strand and thus are transcribed into mRNA [15]. Others are associated with telomeres [16]or oncogene promoters [17]. In the maize genome, G4s tend to occur just downstream of TSSes in the antisense strand (called “A5U”-type G4s for antisense5’-untranslated region) and putative G4s are overrepresented in promoter regions of genes associated with energy status pathways, oxidative stress response, and hypoxia, suggesting a regulatory role for these elements [18]. For these reasons, G4 aptamer-based therapeutics that can inhibit bacteria-host cell interactions [19], override transcriptional [20]or epigenetic [21]signals, or regulate viral lifecycle [22]are an exciting avenue for drug discovery [23].

[12]      Bartas, M. et al. (2019). The Presence and Localization of G-Quadruplex Forming Sequences in the Domain of Bacteria. Molecules 24

[13]      Ruggiero, E., Tassinari, M., Perrone, R., Nadai, M. and Richter, S.N. (2019). The Long Terminal Repeat promoter of Retroviruses contains stable and conserved G-quadruplexes. ACS Infect Dis 

[14]      Scalabrin, M. et al. (2017). The cellular protein hnRNP A2/B1 enhances HIV-1 transcription by unfolding LTR promoter G-quadruplexes. Sci Rep 7, 45244.

[15]      Smestad, J.A. and Maher, L.J. (2015). Relationships between putative G-quadruplex-forming sequences, RecQ helicases, and transcription. BMC Med Genet 16, 91.

[16]      Safa, L., Delagoutte, E., Petruseva, I., Alberti, P., Lavrik, O., Riou, J.-F. and Saintomé, C. (2014). Binding polarity of RPA to telomeric sequences and influence of G-quadruplex stability. Biochimie 103, 80-8.

[17]      Eddy, J. and Maizels, N. (2006). Gene function correlates with potential for G4 DNA formation in the human genome. Nucleic Acids Res 34, 3887-96.

[18]      Andorf, C.M., Kopylov, M., Dobbs, D., Koch, K.E., Stroupe, M.E., Lawrence, C.J. and Bass, H.W. (2014). G-quadruplex (G4) motifs in the maize (Zea mays L.) genome are enriched at specific locations in thousands of genes coupled to energy status, hypoxia, low sugar, and nutrient deprivation. J Genet Genomics 41, 627-47.

[19]      Kalra, P., Mishra, S.K., Kaur, S., Kumar, A., Prasad, H.K., Sharma, T.K. and Tyagi, J.S. (2018). G-Quadruplex-Forming DNA Aptamers Inhibit the DNA-Binding Function of HupB and Mycobacterium tuberculosis Entry into Host Cells. Mol Ther Nucleic Acids 13, 99-109.

[20]      Verma, A., Yadav, V.K., Basundra, R., Kumar, A. and Chowdhury, S. (2009). Evidence of genome-wide G4 DNA-mediated gene expression in human cancer cells. Nucleic Acids Res 37, 4194-204.

[21]      Sengupta, A., Ganguly, A. and Chowdhury, S. (2019). Promise of G-Quadruplex Structure Binding Ligands as Epigenetic Modifiers with Anti-Cancer Effects. Molecules 24

[22]      Métifiot, M., Amrane, S., Litvak, S. and Andreola, M.L. (2014). G-quadruplexes in viruses: function and potential therapeutic applications. Nucleic Acids Res 42, 12352-66.

[23]      Ruggiero, E. and Richter, S.N. (2018). G-quadruplexes and G-quadruplex ligands: targets and tools in antiviral therapy. Nucleic Acids Res 46, 3270-3283.

And adding this sentence to Introduction paragraph 3 addressed point 1, specifically (page 2 lines 55-56)

Some viral genomes combine Watson-Crick base pairs with a G4 structure, perhaps influencing gene expression in HIV-1 [25].

[25]Amrane, S., Kerkour, A., Bedrat, A., Vialet, B., Andreola, M.L. and Mergny, J.L. (2014). Topology of a DNA G-quadruplex structure formed in the HIV-1 promoter: a potential target for anti-HIV drug development. J Am Chem Soc 136, 5249-52.

2) When describing the overall G4 conformations of the locked variants of trim_A5U, it is reported (LL. 237-238 and 254-255) that “A5UAE, A5UAF, A5UAG, A5UBFand A5UBG form antiparallel G4s—major peak at 292 nm”. Indeed the CD spectra have the typical features of hybrid G4s. Please check this.

Thank you for correcting our oversight. We clarified the terminology we used for the antiparallel hybrid G4, as defined in Villar-Geuerra, Gray, and Chaires Curr Protoc Nucleic Acid Chem.2017 Mar 2; 68: 17.8.1–17.8.16.

3) When describing the overall G4 conformations of the locked variants of trim_A5U, it is reported (LL. 237-238 and 254) systematically gonucleotides and their interaction with ZmNDPK1
in both cases carried out with low resolution techniques (CD, UV, DMS footprinting, cryo-TEM), which only give an overall picture of the conformational behavior of the investigated molecules. A systematic characterization would involve a much more detailed study!

We removed that language from the manuscript.

4) When in ll. 359-362 “A5UR20 oligonucleotide, that has 20 random non- G bases followed by a GGGAGGG hairpin on the 3' end, does not have a characteristic G4 TDS spectrum (Figure 5B), but it has parallel G4 CD spectrum in the presence of cation (Figure 5C). We conclude that hex4_A5U forms an intramolecular parallel G4 only in the presence of K+, whereas it forms a weak intermolecular G4 in the presence of Na+, Li+ or Cs+” Indeed dimeric or multimeric aggregates could form in all the studied sequences and the authors should have checked in advance if only monomolecular or also bi/multimolecular G4s are formed under the studied conditions.

We were also concerned about the possibility of oligomerization, which is why we performed the analytical ultracentrifugation experiment (Figure 2). The result show that the oligonucleotide folds into a compact structure with an average molecular weight of 10.8 kDa, close to the expected weight of 10.2 kDa. To emphasize this point, we added to our description of the AUC results (page 4, line 144), which now reads:

Finally, analytical ultracentrifugation (AUC) showed that the DNA was folded into a compact globular structure with an average molecular weight of 10.8 kDa (expected 10.2 kDa) and average of f/f0 of 1.56, indicative of a monomeric G4 (Figure 2).

In addition, a list of few minor errors (typos, etc.) should be fixed before publication, i.e.: 

Thank you for such careful reading of our manuscript. We have corrected the listed typographical errors.

Reviewer 3 Report

General

This is a well-written, scientifically sound manuscript on emerging role of unusual structures DNA can adopt. The main problem with the evaluation of its suitability for publication in Molecules is its novelty relative to the paper of Kopylov et al published in Biochemistry in 2015 (PMID 25679041). The authors should present arguments that this manuscript contains data warranting publication as a full-length paper in Molecules. They showed that there were differences in NDPK binding by folded and unfolded G4 – now they show that similar difference occur when G4 is bulged and non-bulged (canonical) that just confirms a general conclusion on structural-dependence of G4 binding by NDPK.

An apparent advantage of this work is to use a series of G4-forming oligonucleotides with different sequences and, in consequences, different physical-chemical properties, which allows to hypothesize on importance of mutations/mutants in NDPK binding.

Detailed

Affiliation 1 seems to be incomplete

Abstract

“extensive polymorphic G-quadruplex conformations” – do the authors mean many different conformations?

Introduction – in general is too long and para 5th could be skipped or summarized with a single sentence

The 1st sequence can be skipped

“there is no three dimensional structure of an NDPK:G4 DNA complex” – this is somehow misleading – observed, reported?

Results

Results are obtained with suitable methods, are clearly presented and adequately described.

Discussion

It is well-written

Line 351 K → K+

Author Response

Reviewer 3:

This is a well-written, scientifically sound manuscript on emerging role of unusual structures DNA can adopt. The main problem with the evaluation of its suitability for publication in Molecules is its novelty relative to the paper of Kopylov et al published in Biochemistry in 2015 (PMID 25679041). The authors should present arguments that this manuscript contains data warranting publication as a full-length paper in Molecules. They showed that there were differences in NDPK binding by folded and unfolded G4 – now they show that similar difference occur when G4 is bulged and non-bulged (canonical) that just confirms a general conclusion on structural-dependence of G4 binding by NDPK. An apparent advantage of this work is to use a series of G4-forming oligonucleotides with different sequences and, in consequences, different physical-chemical properties, which allows to hypothesize on importance of mutations/mutants in NDPK binding.

The 2015 publication reported the identification of ZmNDPK1 as the first identified plant G4 binding protein from a high-throughput screen for G4 binding proteins in Zea mays, measured its binding constant under a variety of conditions, and determined its X-ray crystal structure. In this publication, we describe the polymorphic nature of the G4 oligonucleotide hex4_A5U and their influence on ZmNDPK1 binding. Additionally, we report for the first time that the ZmNDPK1-G4 protein:nucleic acid complex forms a heterogeneous filament, which in part explains why there is no structure to date of a nucleic-acid bound NDPK, despite our and other’s efforts (for example, Dexheimer, et al, Mol Cancer Therap. 2009). In short, this is a novel body of experimentation that describes phenomena that were not part of the prior study and thus the data warrants publication in Molecules.

Detailed

Affiliation 1 seems to be incomplete

Thank you for noticing – this has now been corrected.

Abstract

“extensive polymorphic G-quadruplex conformations” – do the authors mean many different conformations?

Yes. The language has been clarified and now reads “This sequence adopted an extensively polymorphic G-quadruplex”

Introduction – in general is too long and para 5th could be skipped or summarized with a single sentence

Thank you for the suggestion, we have implemented this change.

The 1st sequence can be skipped

I am afraid I am unsure what the reviewer is suggesting. We feel that Figure 1A is important to include as it explains our nomenclature for defining the stretches of Gs.

“there is no three dimensional structure of an NDPK:G4 DNA complex” – this is somehow misleading – observed, reported?

Thank you for the suggestion, we have implemented it and the sentence (page 2 lines 59-60)) now reads:

“However, the nature of their interaction is not completely understood because there is no reported three dimensional structure of an NDPK:G4 DNA complex.”

Results

Results are obtained with suitable methods, are clearly presented and adequately described.

Discussion

It is well-written

Line 351 K → K+

Thank you for your careful reading of the manuscript, we have made this correction.

Round  2

Reviewer 1 Report

Authors have emphasized some points in the manuscript to address my comment (comment 1). However, there is one point remains doubt. The native sequence adopted multiple G4 structures. The authors want to use mutant sequences to simplified the multiple conformation. However, there are too much different between the mutant and native sequences (both in length and sequence). The authors still could not confirm that the G4 structures by the mutant sequences formed by the native sequence. They should provide smoe commnents and discussions.

Author Response

Thank you for allowing us to further clarify our manuscript. We added yet more clarification to the discussion on page 15, ll 410-413:

In this case, the locked variants describe only one possibility of the variations that the native DNA sequence might adopt. We interpret the data measured on the locked variants to inform us about the ensemble of structures that can possibly form by the native sequence but it could also be that no single mutation exactly mimics the behavior of the oligonucleotide with the full-length, native hex4_A5U sequence.